# Integrating Object View Conditions for Image Synthesis

## Abstract

In the field of image processing, applying intricate semantic modifications within existing images remains an enduring challenge. This paper introduces a pioneering framework that integrates viewpoint information to enhance the control of image editing tasks. By surveying existing object editing methodologies, we distill three essential criteria, consistency, controllability, and harmony, that should be met for an image editing method. In contrast to previous approaches, our method takes the lead in satisfying all three requirements for addressing the challenge of image synthesis. Through comprehensive experiments, encompassing both quantitative assessments and qualitative comparisons with contemporary state-of-the-art methods, we present compelling evidence of our framework's superior performance across multiple dimensions. This work establishes a promising avenue for advancing image synthesis techniques and empowering precise object modifications while preserving the visual coherence of the entire composition. The code will be released.

## 1 Introduction

Applying intricate semantic modifications to existing images is a longstanding and fascinating endeavor within the realm of image processing. The primary objective of image manipulation is to synthesize an image that retains the majority of the existing semantic content while altering specific elements within the source image. In recent years, the landscape of image-to-image models has witnessed a proliferation of methodologies, spanning the spectrum of Generative Adversarial Network (GAN)-based and diffusion-based approaches, encompassing both zero-shot and fine-tuned strategies, all dedicated to addressing this complex task. Faced with this multitude of approaches, a natural inquiry arises: does a given method genuinely fulfill the requirements of precise object modification, and by which criteria is a commendable solution for entity manipulation characterized?

To answer the question, we investigate various image editing applications and make some observations. First, an excellent framework for object modification needs to satisfy the consistency of both the shape and color of the object. Approaches such as Paint-by-Example (Yang et al., 2023) and Paint-by-Sketch (Kim et al., 2023), wherein a reference image is utilized as input for the CLIP model, unfortunately falter in maintaining this object consistency. Conversely, DreamBooth (Ruiz et al., 2023) and its successors (Kumari et al., 2023), exhibit competence in synthesizing objects while preserving their shape and color. Nevertheless, these approaches remain challenged in terms of precise synthesis concerning the object's spatial position and orientation, making them difficult to apply in entity replacement.

Second, in the pursuit of image editing tasks, despite the presence of textual or visual guidance, numerous intricacies often evade direct control and depend on random seed values. For instance, variables such as the precise position and orientation of the synthesis object tend to exhibit a propensity for stochastic occurrence. The issue of object position during synthesis can be effectively mitigated through the application of bounding box constraints, as exemplified by GLIGEN (Li et al., 2023), but with bounding box constraints alone, the object's synthesis remains location-specific without specifying its orientation. Recently, ControlCom (Zhang et al., 2023a), PHD (Zhang et al., 2023b), AnyDoor (Chen et al., 2023), and DreamPaint (Seyfioglu et al., 2023) have also made significant advancements in consistency and controllability. However, without prior reference to the object's

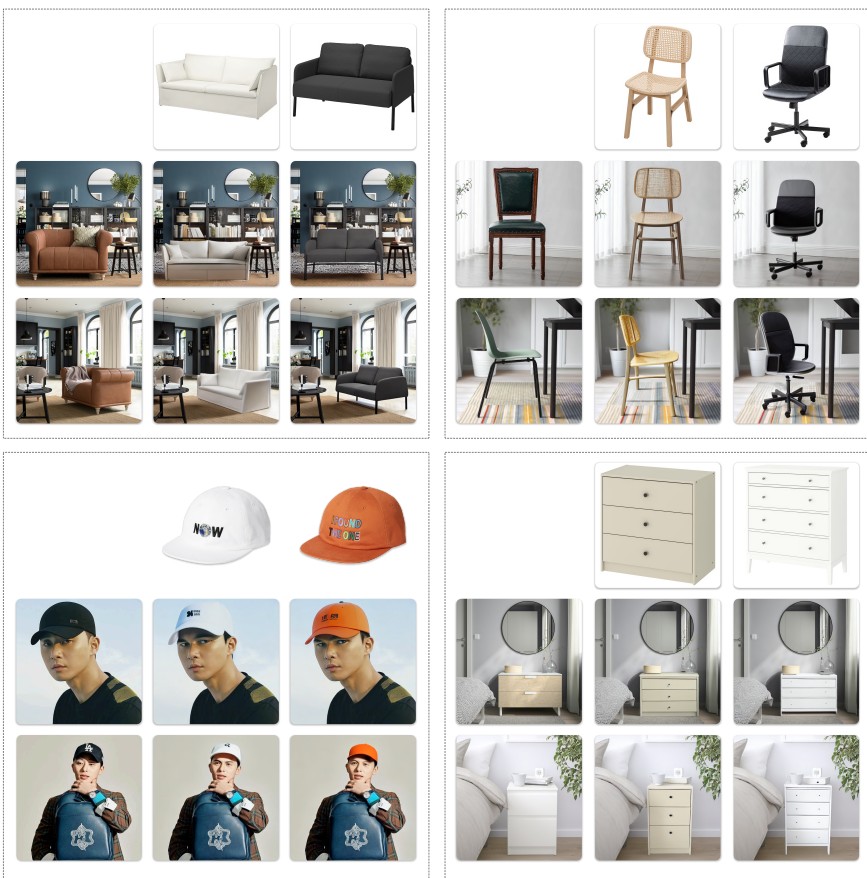

Figure 1: Applications of our proposed method. Our method can replace the object in the left column with the one in the upper row, ensuring not only consistency in the synthesized object but also, by introducing view conditions to the model, enabling precise control over the object's pose and thus enhancing visual harmony.

Table 1: Evaluation Criteria for Image Synthesis Methods: Consistency, Controllability, and Harmony. Consistency refers to the synthesized object being consistent with the reference object. Controllability refers to the ability to manipulate the shape, color, angle, and position of the synthesized object through input. Harmony refers to the coherence between the synthesized object and the original image in terms of lighting, shadow, angle, and positional logic.

| Aspect | PBE | DreamBooth | ControlNet | GLIGEN | ViewControl (Ours) |
|---|---|---|---|---|---|
| Consistency | | ✓ | | | ✓ |
| Controllability | ✓ | | ✓ | ✓ | ✓ |
| Harmony | | ✓ | ✓ | | ✓ |

corresponding camera view, synthesizing specific object directions remains a persistent challenge even given a bounding box.

Third, the resulting synthesis must meet certain quality standards, characterized by harmony in terms of illumination, shading, and logical consistency. Concerning illumination and shading, it is vital that the shadow cast by the synthesized object conforms to the prevailing directional cues within the image. And the reflections displayed by the synthesized object should harmonize with its intrinsic attributes. Furthermore, logical consistency encompasses aspects such as the object's angle, position, and quantity. In summary, the synthesized object must be harmoniously integrated with its surroundings, thereby establishing an optimal state of coordination.

This paper presents a novel framework that enhances existing models with awareness of viewpoint information, thereby enabling improved control over text-to-image diffusion models, such as Stable

Diffusion. This advancement leads to a more controllable approach for image editing tasks. Our proposed pipeline aptly meets all the previously mentioned requirements, with a particular focus on the aspect of controlled pose adjustment, as detailed in Table 1.

To comprehensively evaluate our framework, we assess its performance across various applications, including entity replacement and angle adjustments. This comprehensive evaluation encompasses a wide range of scenarios, such as virtual try-on and interior home design. Notably, we demonstrate that our method yields favorable results across multiple dimensions, even in cases where extensive training is not a prerequisite.

## 2 RELATED WORK

In this section, we first introduce the work related to consistency 2.1, controllability 2.2, and harmony 2.3, and then introduce the work related to novel pose synthesis 2.4.

### 2.1 FEW SHOT PERSONALIZATION AND CUSTOMIZATION

In the context of utilizing just a few reference images, several methods have been proposed to grasp the underlying concept, whether it's a particular theme, style, object, or character. These methods include LoRA (Hu et al., 2021), DreamBooth (Ruiz et al., 2023), Textual Inversion (Gal et al., 2022), HyperNetworks (Ha et al., 2016), and their successors. While these methods and their combinations have opened up avenues for personalized or customized applications with minimal training data, they still rely on having multiple images at their disposal, making it challenging for them to envision different perspectives of an object using just a single image. Furthermore, achieving fine-grained, angle-controllable generation remains a formidable task for them.

Few shot personalization and customization technologies will be popular in e-commerce, because every product in e-commerce catalogs typically has multiple images taken from different angles, which is naturally a good fit for these approaches.

### 2.2 CONDITIONAL AND CONTROLLABLE IMAGE EDITING AND GENERATION

Image-to-image translation (Isola et al., 2017) is a kind of image-conditioned image synthesis, which has been instrumental in image editing and generation, allowing for the preservation of most of the existing semantic content while making specific alterations to particular elements within the source image. These elements can be categorized as style, object, background, and more.

In terms of style, style transfer techniques have played a significant role in advancing controllable image editing and generation. Initially rooted in artistic style transfer, neural style transfer methods, as demonstrated by Gatys et al. (2015), have evolved to grant users greater control over the degree of stylization and the independent manipulation of content and style (Johnson et al., 2016). These developments have facilitated more controlled artistic transformations.

More recently, diffusion models (Ho et al., 2020; Ho & Salimans, 2022) have emerged as the new state-of-the-art family of deep generative models. Representative models such as Stable Diffusion (Rombach et al., 2022b), yield impressive performance on conditional image generation, enabling control over various aspects of the generated content, surpassing those GAN-based (Goodfellow et al., 2020) methods which dominated the field for the past few years. Diffusion models are equipped to accommodate various conditions, whether in the form of textual (Radford et al., 2021) or visual (Zhang & Agrawala, 2023) inputs, making the process of image editing and generation more controllable. However, none of these approaches offer fine-grained control over certain image details, such as lighting, shadows, and object angles.

### 2.3 IMAGE HARMONIZATION

Image harmonization, as explored in previous work (Tsai et al., 2017), focuses on adjusting the illumination and shading between the foreground and background. While several approaches have succeeded in appearance adjustments, they still struggle to address the geometric inconsistencies that may arise between the foreground and background. To handle issues related to inconsistent camera viewpoints, various methods (Chen & Kae, 2019; Lin et al., 2018) have been proposed to estimate warping parameters for the foreground, aiming for geometric correction. However, these methods typically predict affine or perspective transformations, which may not effectively address

more complex scenarios, such as synthesizing foreground objects with novel views or generalizing them to non-rigid objects like humans or animals.

## 2.4 SINGLE IMAGE TO 3D

Before the emergence of CLIP (Radford et al., 2021) and large-scale 2D diffusion models (Rombach et al., 2022b), the conventional approach involved learning 3D priors using either synthetic 3D data (Chang et al., 2015) or real scans (Reizenstein et al., 2021). Unlike 2D images, 3D data can be represented in various formats and numerous representations.

Zero123 (Liu et al., 2023c) is a view-conditioned 2D diffusion model used to synthesize multiple views for object classes lacking 3D assets. It demonstrates that rich geometric information can be extracted directly from a pre-trained Stable Diffusion model, eliminating the need for additional depth information. Building on this, One-2345 (Liu et al., 2023b) utilizes Zero123 to achieve single-image-to-3D mesh conversion.

## 3 METHOD

Given an input image of dimensions $H$ by $W$, a reference image $\mathbf{x}_r \in \mathbb{R}^{H_r \times W_r \times 3}$ containing the reference object, and a prompt description $\mathbf{c}$ (e.g., "Adjust the hat up 10 degrees" or "Replace the laptop on the desk with an apple" as shown in Fig. 1), our objective is to synthesis an output image $\mathbf{y}$ using the information from $\{\mathbf{x}_s, \mathbf{x}_r, c\}$. The goal is to maintain the visual consistency between the output image $\mathbf{y}$ and the source image $\mathbf{x}_s$, while only modifying the object mentioned in $\mathbf{c}$ by the specified angle. Furthermore, when introducing a new object, it is crucial to harmoniously integrate it with the overall composition of the entire image.

This task is particularly intricate due to several inherent challenges. Firstly, the model needs to comprehend the object in the reference image, capturing both its shape and texture while disregarding background noise. Secondly, it is essential to generate a transformed version of the object (varied pose, size, illumination, etc.) that seamlessly integrates into the source image. Furthermore, the synthesized object must align with the original object's angle as specified in $\mathbf{c}$. Lastly, the model must inpaint the surrounding region of the object to produce a realistic image, ensuring a smooth transition at the merging boundary.

Therefore, we adopt the Divide and Conquer principle[1], and break down this intricate problem into easier sub-problems and solve them one by one, in a divide-and-conquer way. To be specific, we address this challenge by combining various generative models, and our combination model is conditioned on the source image $\mathbf{x}_s$, text prompt $\mathbf{c}$, and reference image $\mathbf{x}_r$.

We mathematically formulate our approach as follows:

$$P(\mathbf{y}|\mathbf{x}_s, \mathbf{c}, \mathbf{x}_r) = P(O_s, A_s|\mathbf{x}_s, \mathbf{c}) \cdot P(A_c|\mathbf{c}, A_s) \cdot P(O_r|\mathbf{x}_r, A_c) \cdot P(\mathbf{y}|\mathbf{x}_s, O_s, O_r)$$

In this equation, we use $O_s$ to represent the object within the source image $\mathbf{x}_s$, $A_s$ to denote the angle of this object in $\mathbf{x}_s$, $O_r$ to signify the reference object in the reference image $\mathbf{x}_r$, and $A_c$ to indicate the specific angle extracted from the text prompt $\mathbf{c}$.

The four probabilistic models on the right side of the equation encompass various essential processes within our framework. These processes include object and angle extraction from the source image, angle extraction from the text prompt (as elaborated in Section 3.1), synthesis of the reference object (as elaborated in Section 3.2), and the ultimate image synthesis procedure (as elaborated in Section 3.3). We will now delve into each of these parts and explain how they are integrated.

## 3.1 LLM PLANNER

With in-context-learning and chain-of-thoughts reasoning capabilities, large language models (LLMs) have demonstrated remarkable proficiency in following natural language instructions and completing real-world tasks (Xie et al., 2023; Peng et al., 2023; Liu et al., 2023a). In the appendix, we provide further details about our LLM Planner.

---

[1]https://lllyasviel.github.io/Style2PaintsResearch/

**Pose Estimation and Synthesis**      **Image Synthesis**

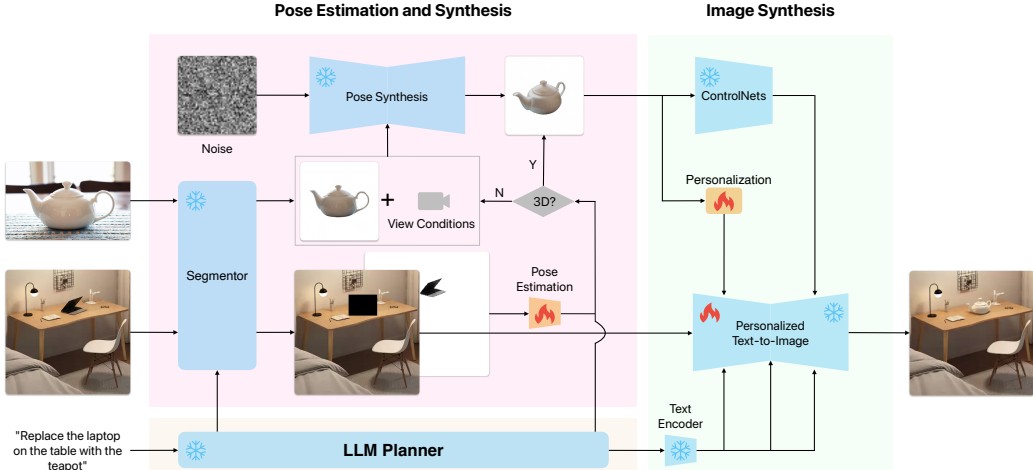

Figure 2: An illustrative overview of our method, which is designed for synthesizing an object with a user-specified view into a scene. "3D?" denotes whether 3D model is available. Our approach consists of three components: Large Language Model (LLM) Planner (Sec. 3.1), Pose Estimation and Synthesis (Sec. 3.2), and Image Synthesis (Sec. 3.3). First, the LLM Planner is adopted to obtain the objects' names and pose information based on the user's input. Second, a segmentation module is adopted to remove the background from the specific object, followed by a pose estimation module to obtain its accurate pose. A pose synthesis module is then applied to synthesize the reference object respecting specific view conditions. Third, a personalized pre-trained diffusion model and ControlNets are adopted to produce the final synthesis. They ensure that the target object harmoniously melds with its surroundings, aligning with the user-specified view, while maintaining consistency in the object's representation. **Flames** and **snowflakes** refer to learnable and frozen parameters, respectively.

## 3.2   POSE ESTIMATION AND SYNTHESIS

In this stage, we present pose representation, pose estimation from a single object image, and the synthesis of an object image given a specific pose.

### 3.2.1   POSE REPRESENTATION

To effectively represent the pose of an object within an image, we employ two fundamental components: the relative camera rotation matrix ($\mathbf{R} \in \mathbb{R}^{3\times3}$) and the relative camera translation vector ($\mathbf{T} \in \mathbb{R}^3$). These elements collectively encapsulate essential information regarding the object's viewpoint and orientation relative to the camera's perspective.

**Relative Camera Rotation ($\mathbf{R}$)**: The matrix $\mathbf{R}$ characterizes the rotation transformation that aligns the object's coordinate system with that of the camera. It encompasses the angular changes required to transition from the object's intrinsic orientation to the camera's frame of reference.

**Relative Camera Translation ($\mathbf{T}$)**: The vector $\mathbf{T}$ denotes the translation in three-dimensional space necessary to position the camera viewpoint with respect to the object. It signifies the displacement along the x, y, and z axes, allowing the object's placement within the scene to be determined.

Together, the relative camera rotation ($\mathbf{R}$) and translation ($\mathbf{T}$) form a comprehensive pose representation, providing a detailed description of the object's spatial orientation and location within the image.

### 3.2.2   POSE ESTIMATION

In this stage, we train a pose estimation model, building upon the foundation of the current image understanding model. The training supervision is

$$\boldsymbol{\Theta} = \arg\min_{\boldsymbol{\Theta}} \mathbb{E}_{\mathbf{x}} \left[ \left\| \hat{\mathbf{R}}_\theta\left(\mathbf{x}\right) - \mathbf{R} \right\|_2^2 + \left\| \hat{\mathbf{T}}_\theta\left(\mathbf{x}\right) - \mathbf{T} \right\|_2^2 \right]$$

Here, $\mathbf{x}$ represents the image, $\mathbf{R}$ and $\mathbf{T}$ denote the relative camera rotation and translation, respectively. $\Theta$ corresponds to the network parameters of our pose estimation model.

Given an object image, our pose estimation model predicts the corresponding relative camera rotation and translation based on the default camera view.

### 3.2.3 POSE SYNTHESIS

We use a view-conditioned diffusion model, Zero123 (Liu et al., 2023c), to generate multi-view images and corresponding pose images. The input to Zero123 consists of a single RGB image $x \in \mathbb{R}^{H \times W \times 3}$ that encompasses the object requiring alignment, and a relative camera transformation rotation $\mathbf{R} \in \mathbb{R}^{3 \times 3}$ and translation $\mathbf{T} \in \mathbb{R}^3$, which is the viewpoint condition control. The output of Zero123 is a synthesized image $\hat{x}_{\mathbf{R},\mathbf{T}}$ capturing the same object from the perspective defined by the transformed camera view:

$$\hat{x}_{\mathbf{R},\mathbf{T}} = f(x, \mathbf{R}, \mathbf{T})$$

where f denotes the freezing model Zero123.

Constrained by its limited generalization capacity, Zero123 excels primarily in a select few categories. Consequently, given the availability of a reference 3D object, we can directly specify view conditions for the reference object to obtain the corresponding image perspective. All images presented in this paper are synthesized by Zero123.

### 3.3 IMAGE SYNTHESIS

Although pose alignment has been achieved, it's possible that the object in the synthesized reference image may have a different size and position compared to the mask in the source image. Therefore, our initial step is to apply padding, either on the left and right or on the top, to the bounding box region of the object in the synthesized reference image. This ensures that the aspect ratio of the object mask in the synthesized reference image matches that of the mask in the source image. Following this, we resize the region that we just padded, ensuring the resized region aligns precisely with the bounding box part in the source image. As a result, we obtain the reference object image $O_r$.

With a source image $I_s$ containing a bounding box mask of the object to be edited and the reference object image $O_r$ with the corresponding camera view, we employ the personalized Stable Diffusion Inpaint Model, controlled by edge and color information, to synthesize the target image.

Why not simply overlay the synthesized object onto the original image? The reason lies in the fact that synthesized objects are typically not perfect, they may exhibit some degree of deformation or error. Consequently, during the image synthesis process, we can only refer to the synthesized object rather than relying on it entirely.

### 3.4 ALL IN ONE

We integrate all the previously mentioned modules to establish an image synthesis framework that allows for view control , as illustrated in Fig. 2. First, we obtain essential object details via the LLM Planner, including angle and object name. Second, we synthesize an appropriate target object image through pose estimation and synthesis. Finally, we employ off-the-shelf diffusion models and associated plugins to achieve pose-controllable image editing.

## 4 EXPERIMENT

### 4.1 IMPLEMENTATION DETAILS

We utilize the following components in our implementation: GPT-4 OpenAI (2023) as our LLM Planner, Segment-Anything (Kirillov et al., 2023) as our segmentation model, Zero-123 (Liu et al., 2023c) as our pose alignment model, and Stable Diffusion v1.5 (Rombach et al., 2022a) [2] and ControlNet 1.1 (Zhang & Agrawala, 2023) as our synthesis models. Additionally, we have developed a pose estimation model, which is trained on ResNet-50 (He et al., 2015).

---

[2]It's worth noting that while we believe SDXL (Podell et al., 2023) performs better in terms of consistency and harmony, its adoption has been temporarily withheld in the current version due to its limited community support and lack of widespread use. We will move to SDXL once related works are done.

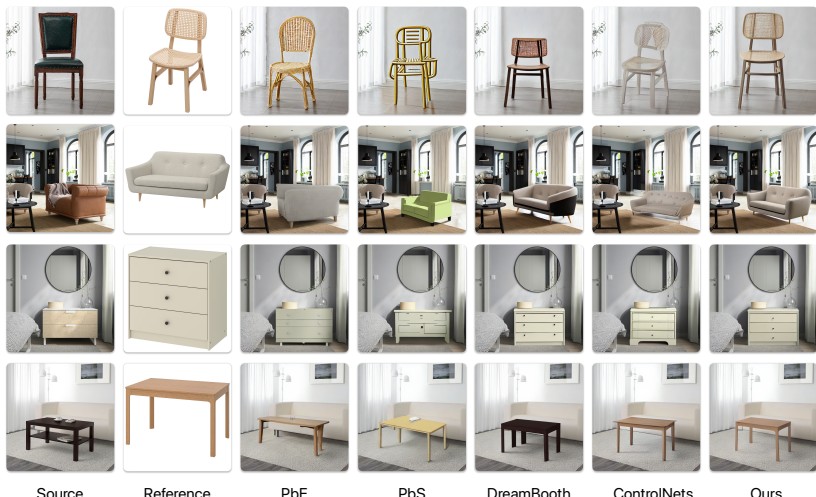

| Source | Reference | PbE | PbS | DreamBooth | ControlNets | Ours |

Figure 3: **Qualitative comparison with reference-based image synthesis methods**, where "PbE" denotes Paint-by-Example (Yang et al., 2023) and "PbS" denotes Paint-by-Sketch (Kim et al., 2023).

In terms of training data, we initially curated product images spanning various categories from publicly available sources on the internet, all captured from a consistent viewpoint, which we have designated as the default camera perspective. Subsequently, employing existing zero-shot novel view synthesis models, we synthesized batches of images, each image batch corresponding to different relative camera viewpoints of each object. In total, our dataset comprises approximately 48.6k images, along with their corresponding relative camera view labels, and we've split them into training and test sets, following an 8:2 ratio. Furthermore, it's important to note that the test set is reserved only for testing.

## 4.2 COMPARISONS

In our comparisons, we have selected recently published open-source state-of-the-art image-driven image editing methods, namely Paint-by-Example (Yang et al., 2023), Paint-by-Sketch (Kim et al., 2023), as our baselines. Figure 3 provides qualitative comparisons and Table 2 provides quantitative comparisons of these methods. We can see that our method consistently achieves superior evaluation results in consistency, harmony and controlliability.

Why don't we employ methods like CLIP Score for quantitative analysis of consistency? Our rationale is rooted in the belief that feature extractors like CLIP often result in the loss of fine-grained image details, which also explains why PbE struggles to achieve consistency. Consequently, evaluating fine-grained generation with a coarse-grained feature extractor may not yield meaningful results. Furthermore, numerous studies have indicated that quantitative evaluation metrics may not consistently align with human perceptual judgments. Given these considerations, we primarily rely on human evaluations to quantitatively assess the performance of our approach and only evaluate the aesthetics score with feature extractors [3].

## 4.3 ABLATION STUDY

In this section, we will begin by discussing the selection process for the pose estimation module backbone, and then demonstrate the essentiality of each component within our image synthesis module. Subsequently, we will show the necessity and robustness of our view conditions. Lastly, we will explain the reason behind our decision not to opt for a two-stage synthesis approach.

### 4.3.1 EFFECTS OF USING DIFFERENT BACKBONES FOR POSE ESTIMATION

We report the prediction error (MAE, mean absolute error) of our pose estimation module with different backbones. And from Table 3, we can see that ResNet-50 achieves better performance with fewer parameters.

---

[3] https://github.com/kenjiqq/aesthetics-scorer

Table 2: **Quantitative Comparisons**. "Consistency" measures the similarity between the reference object and the synthesized object, "Harmony" evaluates the uniformity of pose and view relationships between the background and foreground elements, "Controllability" represents the view information between input and output, and "Aesthetics" denotes the machine evaluation with an aesthetics-scorer. For "Consistency", "Harmony" and "Controllability" evaluation, we collect 15 reviews for each of the 30 sets of synthesized images, with each set comprising three different synthesis methods. Scores were assigned on a scale from 1 to 5, with 1 denoting "terrible", 2 denoting "poor", 3 denoting "average", 4 denoting "good", and 5 denoting "excellent". The aesthetics-scorer will rate each image with an integer range from 1 to 10 (high is good).

| Methods | Consistency ($\uparrow$) | Harmony ($\uparrow$) | Controllability ($\uparrow$) | Aesthetics ($\uparrow$) |
|---|---|---|---|---|
| Paint-by-Example | 2.67 | 2.61 | 1.93 | 4.92 |
| Paint-by-Sketch | 2.79 | 2.21 | 1.87 | 3.93 |
| ViewControl (Ours) | **4.44** | **4.54** | **4.53** | **5.37** |

Table 3: **Quantitative ablation studies on the effects of using different backbones for the pose estimation**, where MAE and RMSE denote mean absolute error and root mean squared error, respectively.

| Methods | #Params | GFLOPs | MAE ($\downarrow$) | RMSE ($\downarrow$) |
|---|---|---|---|---|
| ResNet-50 | 26.20 M | 4.13 G | 4.31 | 7.45 |
| CLIP | 87.88 M | 4.37 G | 3.28 | 10.59 |
| ViT | 86.34 M | 16.86 G | 1.65 | 6.56 |
| DINO-v2 | 85.61 M | 21.96 G | **0.80** | **5.01** |

### 4.3.2 EFFECTS OF IMAGE SYNTHESIS CORE COMPONENTS

We can see from Figure 4 that the components of image generation are indispensable. The personalization module plays a pivotal role in determining the overall object condition, while multiple ControlNets govern the precise object-specific details.

### 4.3.3 EFFECTS OF VIEW CONDITIONS

From Figure 5, we have two key observations:

**Necessity of View Conditions:** In instances where the given view conditions exhibit a significant error or when no view conditions are provided, the process of generating the object tends to favor a semantic orientation within the source image (such as backing against a wall) or the direction most frequently observed during training (typically the front).

**Robustness of View Conditions:** View conditions exhibit a certain degree of robustness. Specifically, predictions remain relatively unaffected by errors within a 20-degree range.

These observations further underscore the dual significance of view conditions, emphasizing both their necessity and robustness.

### 4.3.4 EFFECTS OF 2-STAGE SYNTHESIS

Although a two-stage synthesis approach, involving the initial removal of the original object and subsequent addition of the new object, may mitigate the impact on the original image in certain scenarios, as exemplified by the eyes under the hat in Figure 1. Our framework adheres to more general principles. These principles allow for the possibility of significant disparities in shape between the original object and the new object.

In our experiments, the act of removing the original object often results in the generation of redundant information at the inpaint position. Consequently, when incorporating a new object later, if the mask area for the new object is insufficiently large, this redundant information cannot be effectively eliminated. As a remedy, we employ a larger mask, the bounding box, and opt for a one-stage synthesis approach. Figure 6 visually illustrates such scenarios.

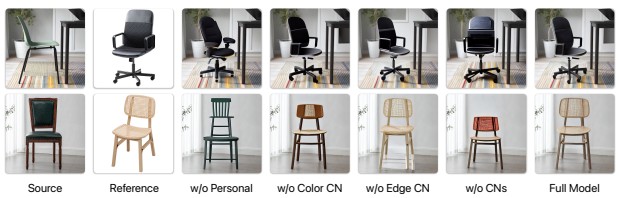

Figure 4: **Qualitative ablation studies on the effects of image synthesis core components**, where "Personal" denotes the personalization module, "Color CN" denotes the ControlNet which controls the color, "Edge CN" denotes the ControlNet which controls the edge, "CNs" denotes all the ControlNets, and "Full Model" denotes with all components.

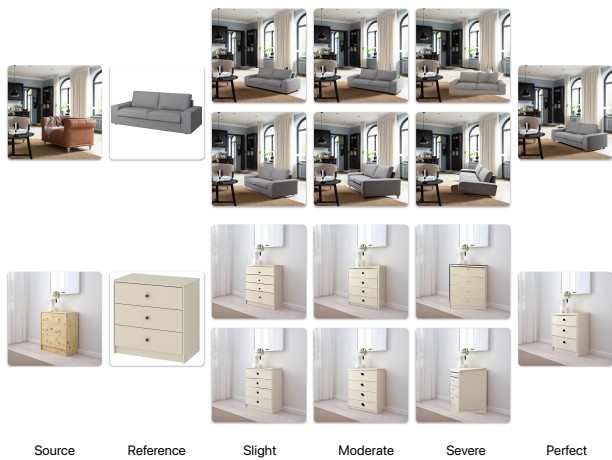

Figure 5: **Qualitative ablation studies on the effects of view conditions**, where "Slight" denotes error range of 0-20 degrees viewing conditions, "Moderate" denotes error range of 20-40 degrees viewing conditions, "Severe" denotes error range of 40-90 degrees viewing conditions, and "Perfect" denotes there is no error.

## 5 CONCLUSION

We present a novel framework that integrates view conditions for image synthesis, which enhances the controllability of image editing tasks. Our framework effectively addresses crucial aspects of image synthesis, including consistency, controllability, and harmony. Through both quantitative and qualitative comparisons with recently published open-source state-of-the-art methods, we have showcased the favorable performance of our approach across various dimensions.

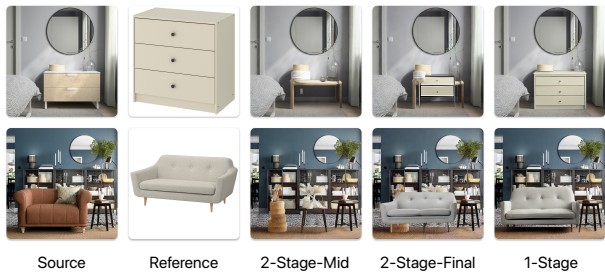

Figure 6: **Qualitative ablation studies on the effects of 2-stage synthesis**, "2-Stage-Mid" denotes the initial inpainting result of the 2-stage synthesis, "2-Stage-Final" denotes the subsequent inpainting result of the 2-stage synthesis, and "1-Stage" denotes the approach that we choose, which involves using only one inpainting step per synthesis.

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

# A   APPENDIX

## A.1   LLM PLANNER

Given the text prompt, our LLM Planner is capable of generating a sequence of commands related to the following operations:

1. **Object Mask and Bounding Box Computation**:
   - **Function**: `compute_mask_and_bounding_box`
   - **Inputs**: `IMG` (Image), `OBJ_NAM` (Object Name)
   - **Outputs**: `MSK` (Mask), `BBOX` (Bounding Box)

2. **Angle Calculation**:
   - **Function**: `calculate_angle`
   - **Inputs**: `IMG` (Image), `MSK` (Mask)
   - **Output**: `ANG` (Angle)

3. **Conditional Angle Calculation**:
   - **Function**: `calculate_conditional_angle`
   - **Inputs**: `ANG` (Angle that needs to be adjusted), `INI_ANG` (Initial Angle)
   - **Output**: `CON_ANG` (Conditional Angle)

4. **Reference Object Synthesis**:
   - **Function**: `generate_reference_object`
   - **Inputs**: `IMG` (Image), `CON_ANG` (Conditional Angle)
   - **Output**: `OBJ_REF` (Reference Object)

5. **Image Synthesis**:
   - **Function**: `generate_image`
   - **Inputs**: `IMG` (Image), `MSK` (Mask), `OBJ_REF` (Reference Object)
   - **Output**: `IMG` (Generated Image)

## A.2   APPLICATIONS

### A.2.1   VIRTUAL TRY-ON

One of the prominent applications of our framework is virtual try-on, which has immense potential in the e-fashion industry. Customers can use our approach to try on different shoes, clothing items, accessories, hats, or even hairstyles virtually. By providing a reference image of themselves and a description like "replace shoes/hats with Object_B," users can see how different fashion items would look on them with specific viewpoint, without physically wearing them. This functionality enhances the online shopping experience, reduces the need for physical trials, and allows for more informed purchase decisions.

Additionally, our framework enables precise adjustments such as "turn Object_A left/right/up/down with 90 degrees," allowing users to customize the placement and orientation of fashion items. This level of control enhances the realism and accuracy of the virtual try-on experience.

### A.2.2   INTERIOR HOME DESIGN

Interior home design is another compelling application of our framework. Users can easily experiment with various furniture and decor options to plan their ideal living spaces. The framework allows users to specify actions like "replace Object_A with Object_B" to switch out furniture items, enabling them to visualize how different pieces would fit into their rooms.

Moreover, users can make detailed adjustments by specifying angles or positions, such as "turn Object_A left/right/up/down with 90 degrees." This level of control ensures that users can fine-tune the placement of furniture and decor items to create harmonious and aesthetically pleasing interiors.

In summary, our framework extends its utility beyond the realm of image editing and synthesis and finds valuable applications in virtual try-on and interior home design. Its ability to handle both object replacement and precise adjustments makes it a versatile tool for various creative and practical tasks.

## A.3  FUTURE WORK

**End-to-End Integration with Latent Space View Control**: While our current approach relies on explicit pose estimation and pose synthesis steps, a promising direction for future work is to further streamline our framework into an end-to-end solution that seamlessly integrates view conditions control within a latent space. This would involve designing a unified model that can adjust view conditions and synthesize different views of objects directly in the latent space. Such latent space view control approach has the potential to significantly enhance inference efficiency and elevate the user experience, making image editing even more accessible and efficient.

**Multi-Object Synthesis**: Extending our framework to support multi-object synthesis is another promising direction for future research. Currently, our approach primarily focuses on modifying a single object while preserving the overall composition of the image. However, many real-world scenarios involve interactions between multiple objects. Enabling the synthesis and manipulation of multiple objects within a scene would open up new possibilities for creative image editing and design. Addressing the complexities of multi-object synthesis, including object-object interactions and spatial arrangements, poses an exciting challenge for future work.

**Video Synthesis**: Another exciting avenue for future research is the extension of our framework to video synthesis. While our current focus is on static images, the demand for dynamic content creation is growing rapidly. Enabling users to apply controlled object modifications and adjustments to video sequences would have applications in panoramic virtual try-on, dynamic interior furniture design, and augmented reality. This expansion would involve addressing temporal coherence and synchronization challenges unique to video synthesis.

In conclusion, our current framework represents a significant step forward in image editing controllability. However, there are exciting opportunities for future work, including the development of end-to-end models with latent space view control, multi-object synthesis, and video synthesis capabilities. These developments aim to significantly enhance inference efficiency and elevate the user experience, providing users with versatile and intuitive tools for creative image and video manipulation, particularly in applications like virtual try-on and interior home design.

## A.4  ADDITIONAL VISUALIZATION RESULTS

In this section, we present four groups of the visualization results showcased in Figures 7, 8, 9, and 10.

The first group aims to validate the effectiveness of various views in reference images, and the effectiveness of synthesizing various views of target images.

The second group aims to validate the effectiveness of handling conflicting conditions, and we provide one successful case and one failed case.

The third group aims to demonstrate the effectiveness of intricate object synthesis and multiple object editing.

The final group provides another visualization result on Mona Lisa portrait.

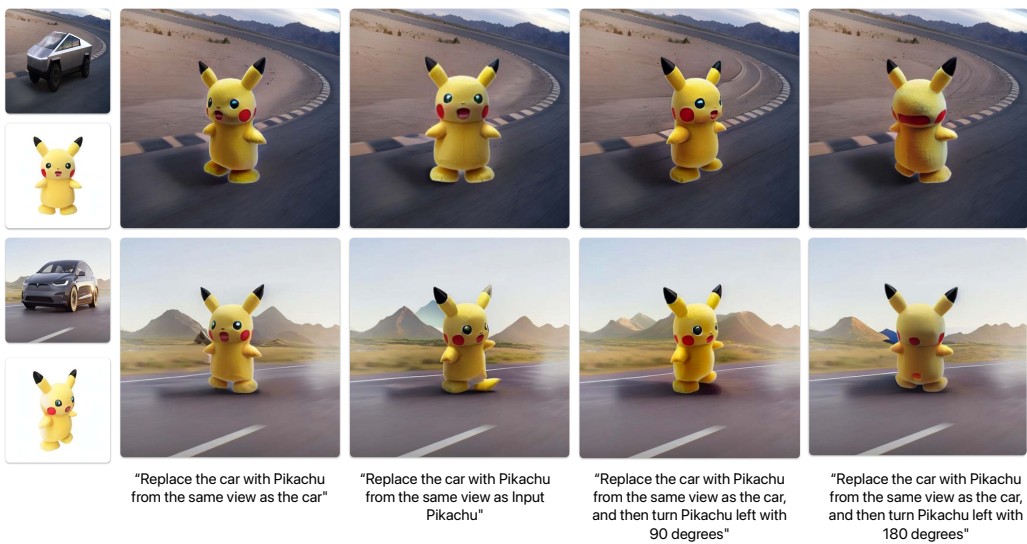

Figure 7: Synthesis of Various Poses of Pikachu

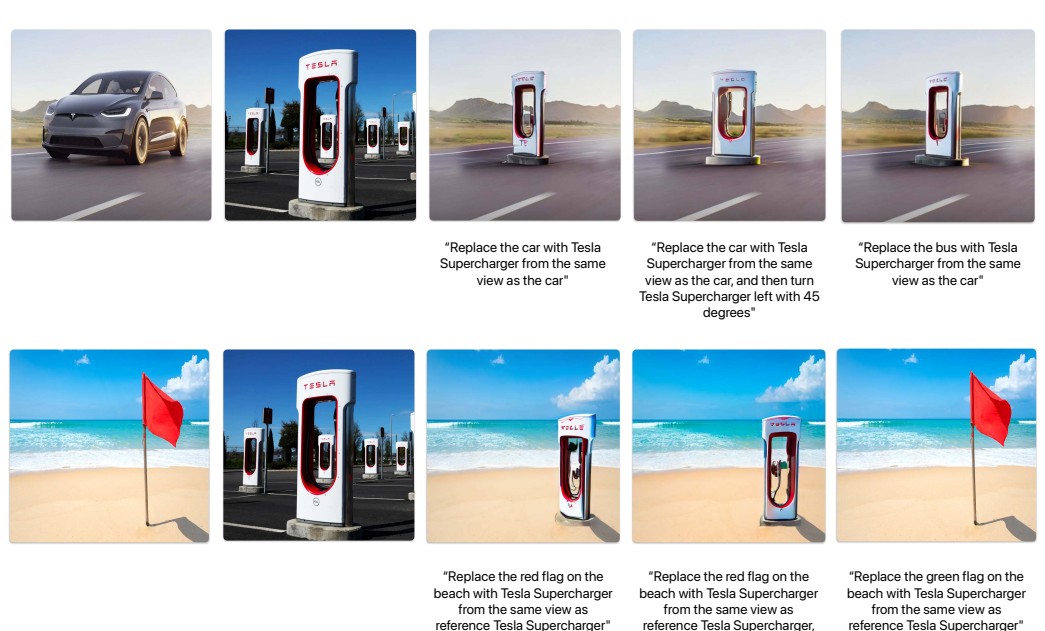

Figure 8: Handling Conflicting Conditions

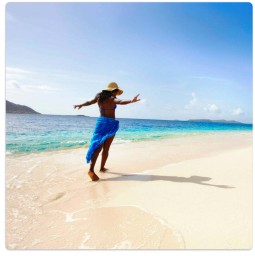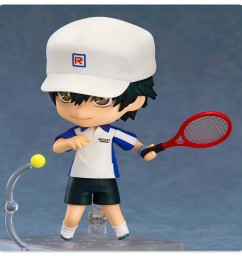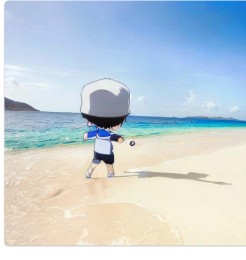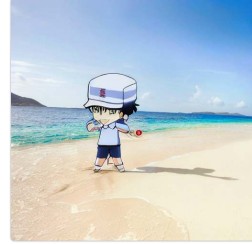

"Replace the girl on the beach with prince of tennis from the same view as the girl"

"Replace the girl on the beach with prince of tennis from the same view as reference prince of tennis"

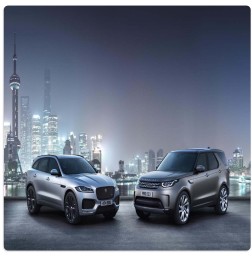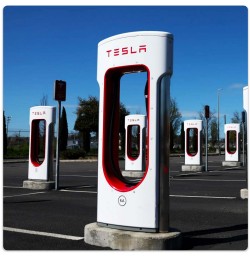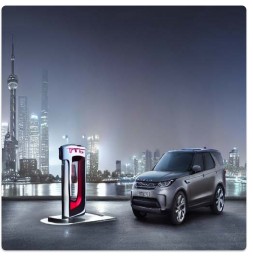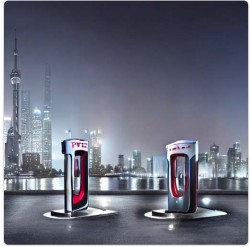

"Replace the right car with Tesla Supercharger"

"Replace the car with Tesla Supercharger"

Figure 9: Intricate Object Synthesis and Multiple Object Editing

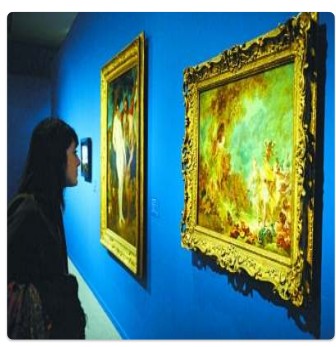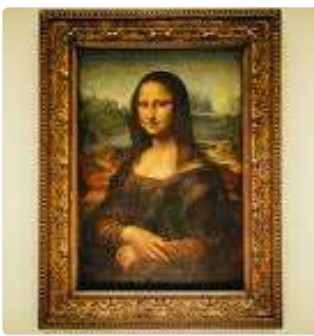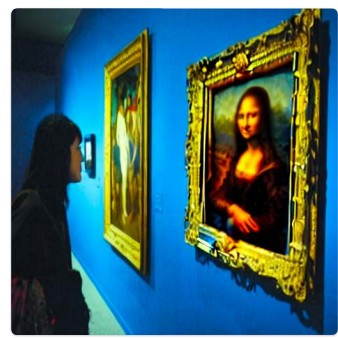

"Replace the right portrait with Mona Lisa portrait"

Figure 10: Visualization Result: Mona Lisa Portrait

