# OpenReview forum: "Integrating View Conditions for Image Synthesis"
_ICLR.cc/2024/Conference — Submitted to ICLR 2024_

### Official Review · Reviewer_RzKy · 2023-10-22

**Soundness:** 3 good
**Presentation:** 3 good
**Contribution:** 3 good
**Rating:** 5
**Confidence:** 5

**Summary:**

The paper introduces an innovative framework that leverages viewpoint information to enhance the controllability of an image editing system. The approach consists of three main components: Large Language Model (LLM) Planner, Pose Estimation and Synthesis, and Image Synthesis.This work provides an approach for object modifications while preserving the visual coherence of the entire composition.

**Strengths:**

The paper is well-written.

The proposed task is both intuitive and has practical applications.

The experiments show compelling performance.

**Weaknesses:**

The lighting estimation, crucial in 3D object integration, solely relies on the diffusion model's generative priors. Considering the complexities distinguishing indoor and outdoor lighting, can the authors provide more outdoor examples?  Specifically, is the model capable of rendering images with realistic lighting effect (diffuse and specular)?  How does its performance compare with a physics-based model on this scene? E.g., cars on streets or buildings during sunset.

Although the paper claims to satisfy the consistency of the object shape, certain details are still modified, e.g., the text on the hat in Fig. 1. While this issue might be intrinsic to Stable Diffusion, the 3D realm demands stricter shape fidelity. Could author provide any improvement for it？

The paper could benefit from showcasing more qualitative results, emphasizing the model's controllability. E.g, how does the model perform when provided with varying textual descriptions to generate a chair in different poses within a living room?

**Questions:**

This framework uses an LLM planner for text guidance. How do the model obtain theses text conditions during testing? Using LLM expand relative camera view labels into sentences?

How does this work address conflicting conditions? E.g., when provided text descriptors (like sofa color or shape) diverge from the original reference, what would be the model's output?

Inserted objects seem basic. Are there any example involving more intricate subjects, e.g., Pikachu, a Picasso painting, or a badminton-playing prince?

I am glad to raise my rating once concerns are addressed.

---

> ### Author Response · Authors · 2023-11-17
> **Response to Reviewer RzKy**
>
> Thank you for your insightful comments. We will appropriately revise some details and add additional visualization results, which is presented as **Appendix A.4 Additional Visualization Results** of the revised PDF now, **and we encourage you to review these to address any remaining concerns you might have**.
>
> **Q**: Can the authors provide more outdoor examples, especially with realistic lighting effects?
>
> **A**: We appreciate the suggestion. In the revised manuscript, we include more outdoor examples to showcase the model's capability to handle diverse lighting conditions, which is presented as **Appendix A.4 Additional Visualization Results** of the revised PDF now. From Appendix 4, we can conclude that based on stable diffusion's ability to understand the entire image, our framework has a certain ability to render images with realistic lighting effect, but it still cannot achieve the same effect as physics-based models.
>
> **Q**: Certain details, like text on the hat in Fig. 1, are modified. Could the authors provide improvements for stricter shape fidelity?
>
> **A**: We acknowledge this concern. This is indeed an issue with Stable Diffusion 1.5. Although there have been partial improvements in SDXL, it is still hard to achieve complete fidelity preservation, even if we provide a very intricate text prompt. In the revision, we'll provide a discussion on this limitation that SDXL performs better but it lacks corresponding community now.
>
> **Q**: Showcase more qualitative results, emphasizing the model's controllability.
>
> **A**: Agreed. We will include additional qualitative results in various scenarios, showcasing the model's controllability and its response to varying textual descriptions for object generation within different contexts. This part is presented as **Appendix A.4 Additional Visualization Results** of the revised PDF now.
>
> **Q**: How does the model obtain view and text conditions during testing?
>
> **A**: On the one hand, LLM Planner will combine the view conditions of the input image, reference image, and those from the input text, calculating the resultant view conditions to be output. These calculated conditions are then fed into the pose synthesis module. On the other hand, LLM Planner will combine Segmentor and BLIP to obtain detailed object description.
>
> There is no need to expand relative camera view labels into sentences, because our pose estimation module accept camera view parameters directly. What LLM needs to do is to pass some numbers to a specific python function.
>
> **Q**: How does the model address conflicting conditions?
>
> **A**: The model's strategy in handling conflicting conditions relies heavily on the configuration of parameters like $text~threshold$ and $box ~threshold$ within the Segmentor component.
>
> When provided text descriptors (like sofa color or shape) diverge from the original reference, the segmentor may struggle to extract relevant information. Consequently, in such scenarios, the resulting output may be the same as the original image. However, the effectiveness of the model in these situations largely relies on the robustness of the segmentor. If the segmentor is robust enough, it can work normally.
>
> We show a successful case and a failed case, and this part is presented as **Appendix A.4 Additional Visualization Results** of the revised PDF now.
>
> **Q**: Are there examples involving more intricate subjects?
>
> **A**: We appreciate the suggestion. Defining the default pose information of intricate subjects like Pikachu, a Picasso painting, or a badminton-playing prince poses a challenge. In such cases, we need collect a series of similar images that given the same view to achieve synthesizing the reference object with the same pose as the source image.
> And view conditions might be invalid due to the absence of a default perspective. Therefore, our framework currently focuses on specific categories, such as indoor furniture, where default perspectives are well-defined.
>
> However, we can also give specific relative view conditions towards the reference images by text prompt, without concerning the view conditions from the source image. And in this case, our model works well. So we also add some examples involving more intricate subjects. This part is presented as **Appendix A.4 Additional Visualization Results** of the revised PDF now.

---

> > ### Comment · Reviewer_RzKy · 2023-11-21
> >
> > Thank you for the authors' response.
> >
> > After the discussion, I believe there still exists potential for further improvement.
> >
> > I have decided to maintain my current rating.

---

### Official Review · Reviewer_j7Dh · 2023-10-31

**Soundness:** 2 fair
**Presentation:** 2 fair
**Contribution:** 2 fair
**Rating:** 5
**Confidence:** 3

**Summary:**

The paper aims to improve the performance of reference-based image editing by explicitly integrating viewpoint information.

The main contribution is a novel system that combines LLM, diffusion models and a camera pose estimation network to achieve high-quality image editing results.

**Strengths:**

1. The idea of leveraging viewpoint information for reference-based object editing in images is novel and interesting.

2. The result shown in the paper looks promising, and the improvement over the SOTA is obvious.

**Weaknesses:**

1. The scale of technical novelty is limited. The whole system looks like an ensemble of various existing approaches (e.g., GPT-4, Segment-Anything, Zero-123, and ControlNet), with any significant modification to them. While I do acknowledge the engineering effort that the authors have put into carefully selecting proper components and figuring out a reasonable way to compose them together, the novelty of the paper, from a technical perspective, is insufficient for ICLR.

2. The evaluation is incomplete. First, the results shown throughout the paper are all on indoor scenes. An experiment should be done to test the ability of the proposed method in handling outdoor images, which is not included in the current paper. Second, only two out of the three dimensions including consistency, controllability and harmony are considered in the experiments, as shown in Table 2.  The controllability of different methods should also be tested, e.g., through a human evaluation where subjects can be asked to tell how well the pose of the synthesized object aligns with that of the corresponding object in the source image.

**Questions:**

1. In Fig.2, what is the personalized pre-trained diffusion model? What is the trainable “personalization” module and how is it trained?

2. Can the system work well if there are more than one object to be replaced in the source image? For example, what about if the source image in Fig. 2 contains two laptops on the table?

3. What dataset is used for the comparisons in Sec. 4.2？

---

> ### Author Response · Authors · 2023-11-17
> **Response to Reviewer j7Dh**
>
> Thank you for your constructive comments. We will appropriately revise the evaluation table and add additional visualization results, which is presented as **Appendix A.4 Additional Visualization Results** of the revised PDF now, **and we encourage you to review these to address any remaining concerns you might have**.
>
> **Q**: Why integrate view conditions for image synthesis. What is technically unique and novel in it?
>
> **A**: While drawing upon established methodologies, our unique innovation rests in the integration of view conditions for object synthesis. This novel approach presents a **distinct** and **pragmatic** solution to a prevalent challenge in image synthesis. The strength of our framework lies in its adaptability across diverse domains while requiring minimal training resources (e.g., 1 GPU for 6 hours), setting it apart from resource-intensive alternatives (e.g., a concurrent method that demands 8 GPUs for 144 hours). We underscore the novelty within the synthesis framework itself, rather than focusing solely on individual components.
>
> Our system's innovation stems from the fusion of established components, specifically designed for viewpoint-guided image editing. The uniqueness lies in our methodological approach, a novel combination and application of these components, specifically tailored to tackle the complexities of viewpoint-based image synthesis. Similar to the best paper "Planning-oriented Autonomous Driving" showcased at CVPR 2023, our method extends and refines existing methodologies to effectively address the practical challenges in this domain.
>
> **Q**: Incomplete evaluation.
>
> **A**: We appreciate your feedback regarding the "Incomplete evaluation". In response, our revised version will feature additional experiments involving outdoor scenes, aiming to illustrate the versatility of our method. Besides, it's essential to note that our method primarily focuses on modifying specific objects under defined view conditions, irrespective of the background. Hence, the distinction between indoor and outdoor scenes may not significantly impact our core findings.
>
> Furthermore, while we have outlined an evaluation of controllability in Sec. 4.3.3, we acknowledge the need for a more comprehensive assessment by human evaluation. To address this, we conduct a human evaluation where evaluators will judge the alignment of synthesized object poses with input conditions. The resulting scores from this evaluation will be incorporated into the revised manuscript to provide a more detailed and comprehensive evaluation of controllability.
>
> | Methods         | Consistency ($\uparrow$) | Harmony ($\uparrow$) | Controllability ($\uparrow$) | Aesthetics ($\uparrow$) |
> |-----------------|---------------------------|-----------------------|------------------------------|--------------------------|
> | Paint-by-Example| 2.67                      | 2.61                  | 1.93                         | 4.92                     |
> | Paint-by-Sketch | 2.79                      | 2.21                  | 1.87                         | 3.93                     |
> | ViewControl (Ours) | **4.44**               | **4.54**              | **4.53**                     | **5.37**                 |
>
> **Q**: Details about personalization module.
>
> **A**: In Fig. 2, the personalized pre-trained diffusion model denotes a model fine-tuned to accommodate unique object characteristics, such as Dreambooth, LoRA, and their successive versions. The personalization module represents some trainable parameters, and it is optimized by images acquired throughout the pose synthesis process.
>
> The specific method utilized within this personalization module is not explicitly detailed due to the rapidly development of this domain. The selection of method for personalization should be guided by practical applications, aiming to find an optimal balance between efficiency and performance. For example, if your want to preserve more details of the reference object, Dreambooth is a good choice. But if you need faster training speed, LoRA might be a better choice.
>
>
> **Q**: Can the system work well if there are more than one object to be replaced in the source image?
>
> **A**: Yes, this is equivalent to going through the process multiple times, processing only one object each time. And our system can sucessfully handle it, seen as **Appendix A.4 Additional Visualization Results** of the revised PDF. This will work well when the Segmentor module works well.
>
> **Q**: What dataset is used for the comparisons in Sec. 4.2?
>
> **A**: We used the dataset comprised of product images spanning various categories, gathered by our team from publicly available repositories, as described in Sec. 4.1 of our manuscript. It's important to note that we strictly maintain a clear distinction between the training set and the test set, ensuring that the data used in comparisons have not been part of the training process.

---

> > ### Comment · Reviewer_j7Dh · 2023-11-23
> >
> > I appreciate the authors' detailed responses to my comments, but I am still concerned with the lack of comprehensive experiments on outdoor examples and the amount of technical originality on combining previously established components. Hence, I decide to keep my original rating.

---

### Official Review · Reviewer_pMRb · 2023-11-01

**Soundness:** 2 fair
**Presentation:** 2 fair
**Contribution:** 2 fair
**Rating:** 5
**Confidence:** 3

**Summary:**

The paper introduces a method for inserting an object into a scene based on a user-defined view. This method consists of three steps: First, using the Large Language Model (LLM) Planner to identify the object and its pose from user input. then, using a segmentation network to isolate the object and determine its pose, followed by pose synthesis that respects user-specified views. Finally, using a personalized diffusion model and ControlNets for the final image synthesis, ensuring the object blends naturally with the scene. This process aims to improve the composition effects using precise pose estimation and optimal view conditions.

**Strengths:**

The paper treads on a fresh ground by presenting a new and practical problem setting, which always adds value by opening up new directions for research. By merging existing works, the paper provides a comprehensive approach to tackle the problem.

**Weaknesses:**

1. While the problem setting is fresh, the technique to address it lacks originality, relying heavily on previously established works without introducing significant innovative components or showing strong technical contributions.
2. The proposed pose estimation method, based on the relative camera rotation matrix and translation vector, is rudimentary. Its simplistic model might effect its generalization in a practical settings. For example in figure 1 and 3, all the reference photos are in roughly the same orientations (e.g. 30 degrees left forward facing) and the source objects are either in the same orientations with the reference or forward facing directions. The paper might not have sufficiently demonstrated the effectiveness of their method in diverse scenarios.

**Questions:**

1. Considering the similar orientations in figures 1 and 3, how does the method perform with objects in diverse orientations? Would it be possible to provide more challenging examples or demonstrate the same object transformed under various orientations and scales?
2. the performance of pose estimation shows a RMSE value of 9.7, even in the best scenario as shown in table 3, seems relatively high. Is there room for improvement?
3. In section 'EFFECTS OF VIEW CONDITIONS', suggesting that predictions are marginally affected within a 20-degree range, is perplexing. This statement seems to undermine the paper's central argument, suggesting that pose estimation and view conditions may not be as critical or even necessary. Could the authors elaborate on this?

---

> ### Author Response · Authors · 2023-11-17
> **Response to Reviewer pMRb**
>
> Thank you for your careful comments. We will appropriately revise the pose estimation performance table and add additional visualization results, which is presented as **Appendix A.4 Additional Visualization Results** of the revised PDF now, **and we encourage you to review these to address any remaining concerns you might have**.
>
> **Q**: Why integrate view conditions for image synthesis. What is technically unique and novel in it?
>
> **A**: While drawing upon established methodologies, our unique innovation rests in the integration of view conditions for object synthesis. This novel approach presents a **distinct** and **pragmatic** solution to a prevalent challenge in image synthesis. The strength of our framework lies in its adaptability across diverse domains while requiring minimal training resources (e.g., 1 GPU for 6 hours), setting it apart from resource-intensive alternatives (e.g., a concurrent method that demands 8 GPUs for 144 hours). We underscore the novelty within the synthesis framework itself, rather than focusing solely on individual components.
>
> **Q**: Similar orientations in figures 1 and 3.
>
> **A**: The similar orientations in the reference images depicted in figures 1 and 3 can be attributed to their adherence to the default view predetermined within our pose estimation module. Acknowledging the necessity for a more varied representation, we include a broader array of examples showcasing with objects in diverse orientations, which is presented as Appendix 4 in the final section of the revised PDF now. Specifically, we estimate both the relative view information ($v1$) of the original object and the relative view information ($v2$) of the reference object. With the relative view information ($v3$) provided by user prompt, by computing $|v2-v1+v3|$, we derive the relative view information required to transform the reference object effectively into the view of the original object.
>
> Regarding scale information, if the original objects and reference objects are of the same type, there are no issues. However, when dealing with different types of objects, our approach involves employing a series of traditional transformations to adjust the reference object to the mask area. Consequently, there might be some challenges in open domain data. We leave it for future work, by providing a solution where users can manually scale objects through the graphical user interface (GUI).
>
> **Q**: Details about table 3.
>
> **A**: We appreciate your feedback and have updated Table 3 as follows with more experiments with more suitable hyperparameters to improve accuracy.
>
> | Methods   | \#Params | GFLOPs | MAE ($\downarrow$) | RMSE ($\downarrow$) |
> |-----------|----------|--------|---------------------|---------------------|
> | ResNet-50 | 26.20 M  | 4.13 G | 4.31                | 7.45                |
> | CLIP      | 87.88 M  | 4.37 G | 3.28                | 10.59               |
> | ViT       | 86.34 M  | 16.86 G| 1.65                | 6.56                |
> | DINO-v2   | 85.61 M  | 21.96 G| **0.80**            | **5.01**            |
>
> **Q**: Details about section "Effects of View Conditions".
>
> **A**: We apologize for any confusion arising from our previous explanation regarding the "Effects of View Conditions". Allow us to clarify further. Our initial intent was to highlight the robustness of our predictions within a reasonable variance. Nevertheless, it is crucial to claim the importance of precise pose estimation and accurate view conditions in achieving realistic object synthesis.
>
> The slight column indicates a level of robustness in our predictions because in these tests, they can combine the semantic information of the background and the pose information of the reference object to comprehensively determine the generated pose information. But in many cases, such as empty outdoor scenes, there is no clear background semantic information to guide the pose of generated objects. Besides, when comparing these slight variations to the perfect column, it becomes evident how crucial precise pose estimation and accurate view conditions are in achieving a more accurate outcome in object synthesis. Or in other words, robustness determines the lower bound of the generated effect, and effectiveness determines the upper bound of the generated effect.

---

### Official Review · Reviewer_H34R · 2023-11-07

**Soundness:** 3 good
**Presentation:** 4 excellent
**Contribution:** 3 good
**Rating:** 5
**Confidence:** 4

**Summary:**

This paper introduces a new framework for image synthesis that integrates view conditions to enhance the controllability of image editing tasks. The framework satisfies three essential criteria for an effective image editing method: consistency, controllability, and harmony. The paper surveys existing object editing methodologies and distills these criteria, which should be met for an image editing method. The paper describes the various processes involved in the framework, including object and angle extraction, text prompt processing, reference object synthesis, and image synthesis. The paper also presents evidence of the framework's superior performance across multiple dimensions through comprehensive experiments and comparisons with state-of-the-art methods.

**Strengths:**

- The framework integrates view conditions to enhance the controllability of image editing tasks, allowing for precise object modifications while preserving the visual coherence of the entire composition.

- The framework satisfies three essential criteria for an effective image editing method: consistency, controllability, and harmony.

- The paper presents evidence of the framework's superior performance across multiple dimensions through comprehensive experiments and comparisons with state-of-the-art methods.

**Weaknesses:**

- I think the title of this paper does not really satisfy its content. "View-conditioned image synthesis" normally refers to controlling the camera view of the entire image, but the proposed method actually sounds more like editing local regions with view conditions. It is more like a system design and the current title does not fit it well.

- To put the new object in the original scenes with the same 6D pose, the authors proposed to use a pose estimation module. I have several related questions regarding this part.
    1) Does this pipeline require the new object to be presented in the canonical view? It seems that the pose estimator only produces relative camera parameters.
    2) Why does the generated image in the presented results always follows the 6D pose of input images? What if we choose a different pose? Will it fail the framework?
    3) In most results, the newly added object and the existing object are from the same category. What about different categories?

**Questions:**

N/A

---

> ### Author Response · Authors · 2023-11-17
> **Response to Reviewer H34R**
>
> Thank you for your insightful comments. We will appropriately revise the title and add additional visualization results, which is presented as **Appendix A.4 Additional Visualization Results** of the revised PDF now.
>
> **Q**: Why integrate view conditions for image synthesis?
>
> **A**: The integration of view conditions in image synthesis addresses one limitation in current image editing methods, which often lack precise control over view-related information. It's essential to highlight that, similar to text-guided and image-guided image editing techniques, our approach specifically applies view conditions to the object of interest rather than the entire image. This utilization of view conditions significantly enhances controllability. The title, "View-conditioned Image Synthesis," aligns well with this approach, and we elaborate its significance in the Abstract.
>
> Inspired by diverse text-guided and image-guided regional image editing methodologies, we formulated this title. Your constructive feedback is immensely appreciated, and **we intend to incorporate your suggestions**. Here are some refined title options:
>
> "Integrating Object View Conditions for Image Synthesis"
>
> "Integrating View Conditions for Regional Image Synthesis"
>
> "Integrating View Conditions for Image Editing"
>
> **Q**: Provide more details about the pose estimation module.
>
> **A**: In general, without the view estimation of reference object, our pipeline needs the presentation of the new object in a canonical view. However, we have devised strategies to achieve this without relying explicitly on a canonical view. For instance, we estimate both the relative view information ($v1$) of the original object and the relative view information ($v2$) of the reference object. With the relative view information (v3) provided by user prompt, by computing $|v2-v1+v3|$, we derive the relative view information required to transform the reference object effectively into the view of the original object.
>
> The images generated in the presented results align with both the 6D pose of input images and accompanying text descriptions. While we have showcased only a selection of examples without text descriptions, our revised version will include more diverse instances, such as "Replace A with B, and turn left 90 degrees." Generally, when the new pose harmonizes with the background, the resulting effect tends to be favorable.
>
> In scenarios where the newly added object and the existing object do not belong to the same category, it still works well. Nevertheless, in practical applications, replacing objects across different categories is rare. Our system is tailored to address realistic scenarios, ensuring, for example, that a shoe would not inadvertently substitute for a hat. Despite this, our revised version will include more scenarios where where the newly added object and the existing object do not belong to the same category.
>
> We highly value your feedback and aim to address these concerns comprehensively in the revised manuscript by incorporating additional examples and explanations. **We have included some visualization results in Appendix 4 of the revised PDF, and we encourage you to review these to address any remaining concerns you might have**.

---

### Meta-Review · Area_Chair_7y8n · 2023-12-07

**Metareview:**

This paper proposes a method for enhancing the controllability of image editing by incorporating viewpoint information. The proposed process involves identifying objects and poses, processing text prompts, synthesizing reference objects, and synthesizing images. This system integrates LLM, diffusion models, and a camera pose estimation network. In this way, the proposed approach is capable of modifying and synthesizing objects to preserve the visual coherence of the composition, thereby simplifying the process of editing images.

The proposed framework integrates view conditions to enhance the controllability of image editing tasks, allowing for precise object modifications while maintaining the visual coherence of the composition. Experiments show compelling results and better performance than baseline methods. The proposed method could facilitate the process of editing images more efficiently and intuitively.

There are concerns regarding the technical novelty of the proposed approach since it relies heavily on previously established works without introducing significant innovative components. It has been questioned whether the proposed method requires similar orientation, whether the new object must be presented in the canonical view, and whether the newly added object and the existing object must be of the same category. In addition, the experiments only contain indoor scenes. In the revised paper, additional results demonstrate that these conditions are not necessary. It has also been noted in reviews that certain parts of the results are highly dependent on the diffusion model, such as lighting conditions and details of the objects. The reviewers also note that the paper title does not adequately describe the target task and results presented. There are still concerns about novelty and comprehensive experiments after the discussion stage.

**Justification For Why Not Higher Score:**

Novelty and experiment issues are not resolved even after the discussion stage. The reviews are all slightly negative.

**Justification For Why Not Lower Score:**

N/A

---

### Decision · Program_Chairs · 2024-01-16

Reject